# TGF-β Type II Receptor Punt Suppresses Antimicrobial Peptide Expression and Influences Development in *Tribolium castaneum*

**DOI:** 10.3390/insects14060515

**Published:** 2023-06-02

**Authors:** Jingjing Li, Bo Lyu, Qisheng Song

**Affiliations:** Division of Plant Science and Technology, University of Missouri, Columbia, MO 65211, USA; jli@mail.missouri.edu (J.L.); bl3pt@missouri.edu (B.L.)

**Keywords:** Punt, TGF-β receptor, *Tribolium castaneum*, AMP, immune response, RNAi

## Abstract

**Simple Summary:**

The red flour beetle, *Tribolium castaneum*, is a severe pest that infests stored agricultural products and serves as a widely used model insect for molecular study. While transforming growth factor-β (TGF-β) has been extensively studied in vertebrates, its role in insects remains to be thoroughly investigated. In this study, we report our findings on the impact of TGF-β type II receptor Punt on the antimicrobial peptide (AMP) expression in *T. castaneum*. *Punt* RNAi in the larvae led to increased transcript levels of AMP genes via a transcription factor Relish in the IMD pathway, leading to inhibition of *Escherichia coli* proliferation. Knockdown of *Punt* in larvae also resulted in the splitting of elytra and abnormal compound eye development in adults. When *Punt* was knocked down during the female pupal stage, it not only increased transcript levels of AMP genes but also had adverse effects on ovarian development, leading to a decrease in fecundity and failure of egg hatching.

**Abstract:**

The transforming growth factor-β (TGF-β) superfamily in insects regulated various physiological events, including immune response, growth and development, and metamorphosis. This complex network of signaling pathways involves conserved cell-surface receptors and signaling co-receptors that allow for precisely coordinated cellular events. However, the roles of TGF-β receptors, particularly the type II receptor Punt, in mediating the innate immunity in insects remains unclear. In this study, we used the red flour beetle, *Tribolium castaneum*, as a model species to investigate the role of TGF-β type II receptor Punt in mediating antimicrobial peptide (AMP) expression. Developmental and tissue-specific transcript profiles revealed *Punt* was constitutively expressed throughout development, with the highest transcript level in 1-day female pupae and the lowest transcript level in 18-day larvae. Tissue specific expression profiles showed the highest transcript level of *Punt* was observed in the Malpighian tubule and ovary in 18-day larvae and 1-day female adults, respectively, suggesting Punt might have distinct functions in larvae and adults. Further results indicated that *Punt* RNAi in the 18-day larvae led to increased transcript level of AMP genes through transcription factor Relish, leading to inhibition of *Escherichia coli* proliferation. Knockdown of *Punt* in larvae also led to splitting of adult elytra and abnormal compound eyes. Furthermore, knockdown of *Punt* during the female pupal stage resulted in increased transcript levels of AMP genes, as well as abnormal ovary, reduced fecundity, and failure of eggs to hatch. This study deepens our understanding of the biological significance of Punt in insect TGF-β signaling and lays the groundwork for further research of its role in insect immune response, development, and reproduction.

## 1. Introduction

The transforming growth factor-β (TGF-β) superfamily comprises a diverse group of proteins that includes TGF-β isoforms, bone morphogenetic (BMP) proteins, and activins [1]. The TGF-β family is evolutionarily conserved in metazoans and has been shown to be expressed in germ cells, embryos, and various tissues and organs, regulating diverse cellular behaviors that influence development and immune systems in both vertebrates and invertebrates [2]. The TGF-β signaling pathway is highly conserved in mammals and is known to play crucial roles in inflammation and tissue repair [3]. In the model insect *Drosophila melanogaster*, studies have shown that TGF-β signaling also contributes to the immune response to wounding and bacterial infection [4]. Specifically, the TGF-β ligand decapentaplegic (Dpp) is activated upon wounding, leading to a downregulation of antimicrobial peptides (AMPs). Another TGF-β ligand, Dawdle (Daw), limits infection-induced melanization. Additionally, in the domestic silk moth, *Bombyx mori*, research has demonstrated that the transcript levels of *Daw* and *Dpp* are subject to differential regulation in the hemocytes of larvae infected with the nucleohedrovirus BmNPV. Notably, overexpressing *Daw* and *Dpp* can reduce virus replication, while RNAi knockdown of *Daw* and *Dpp* in *B. mori* culture cells lead to an increase in virus production [5]. Recent research has shed light on the role of *Dpp* and *Daw* in the immune response to nematode infections in fruit flies, indicating that the transcriptional induction of these TGF-β ligands can impact the survival ability of infected flies [6]. Additionally, a study highlights the influence of TGF-β signaling on the interaction between the host red palm weevil, *Rhynchophorus ferrugineus*, and the parasitic nematode *Steinernema carpocapsae* through RNA-seq analysis [7]. However, the precise mechanism underlying this regulation remains unclear. While the impact of TGF-β signaling on the embryonic and larva to adult development of the red flour beetle, *Tribolium castaneum*, has been studied [8,9,10,11], the effect of TGF-β signaling on the immune response of other insects, especially coleopteran insects is still not well understood. Despite these findings contributing significantly, further exploration is needed to elucidate the relationship between TGF-β pathways and insect immune pathways, such as the immune deficiency (IMD), Toll, and Janus kinase/signal transducer and activator of transcription (JAK/STAT) pathways. In comparison, there has been a more substantial body of research focused on the role of TGF-β in insect wing development and reproduction [12,13,14,15,16,17,18,19,20].

Dpp has been established as a significant regulator of growth and patterning in imaginal discs across numerous insect species [12,13,14]. In contrast, while activin signaling is known to play a role in cell proliferation, its involvement in tissue patterning appears to be limited. Furthermore, proper growth and patterning of the wing disc in insects rely on the activity of a second BMP ligand, Gbb [15]. Unlike *Dpp*, *Gbb* is more broadly expressed throughout the developing wing tissue, and its loss leads to wing growth and patterning defects similar to those caused by Dpp loss, although generally less severe [16]. Regarding its action in reproduction, TGF-β signaling plays a role in regulating oogenesis in female insects, and is well-studied in dipteran and lepidopteran, but not in coleopteran. It governs the development of germline and surrounding follicle cells, which are crucial for supporting the developing egg. Loss of Dpp-mediated TGF-β signaling can lead to defects in oogenesis, such as abnormal eggshell formation and defective dorsal-ventral patterning [17,18,19]. Besides its function in gametogenesis, TGF-β signaling is also involved in regulating embryonic development, specifically in patterning the embryonic body axis and governing the development of gut and nervous systems [20].

In *Drosophila*, the TGF-β family of ligands initiate signal transduction by binding to TGF-β receptors on the plasma membrane. These receptors consist of two types: type I and type II, both of which are single-spanning transmembrane proteins that possess a cytoplasmic serine/threonine kinase domain. According to molecular genetics and structural studies, the active receptor complex in TGF-β signaling is a tetramer consisting of two type I and two type II molecules [21]. The type II receptors Punt and Wishful thinking (Wit) are responsible for TGF-β signaling. Punt is widely regarded as the primary Type II receptor, due to its broad expression pattern, severe loss-of-function phenotype, and ability to interact with both BMP and Activin ligands [22]. These specific features make Punt a focal point in insect research. For example, elimination of Punt in the wing disc of *Drosophila* results in the decrease of epithelial cells [23]. Recent studies have highlighted additional roles for Punt in *Drosophila*. Knocking down of *Punt* in salivary glands leads to alterations in nucleolar structure and function [24], underscoring the importance of Punt in maintaining normal cellular processes. Additionally, Punt is essential for Dpp to play a critical role during larval eye development [25]. Punt’s correct basolateral presentation is necessary for optimal TGF-β signaling and normal function; otherwise, *Drosophila* may experience developmental defects, female sterility, and even significant lethality [26].

In 1996, researchers first characterized the TGF-β superfamily in *T. castaneum* in an effort to explore the evolutionary conservation of TGF-β-like genes and their functions during insect development [27]. The subsequent study indicated that *T. castaneum* may have preserved a more primitive TGF-β signaling component composition compared to the *Drosophila* lineage, which underwent significant changes over time. However, the precise roles of TGF-β signaling in mediating immunity and development of *T. castaneum* remain unclear. Nonetheless, *T. castaneum* serves as an excellent insect model for studying the functions of these ancestral signaling components due to its available genome sequence and high efficiency of RNAi [28]. In this study, we primarily investigated the effects of knocking down *Punt* at different development stages of *T. castaneum* on the immunity, wing and eye development, and reproduction of the beetles. Our results showed that *Punt* RNAi in 18-day-old larvae led to upregulated expression of AMP genes through Relish, a NF-κB transcription factor in the IMD signaling pathway [29]. Knockdown of *Punt* in larvae also caused splitting of adult elytra and abnormal compound eyes. Furthermore, knocking down *Punt* during the female pupal stage resulted in increased transcript levels of AMP genes as well as abnormal ovarian development, decreased fecundity, and failure of egg hatching.

## 2. Materials and Methods

### 2.1. Insect Rearing

We conducted all experiments using the *T. castaneum* Georgia-1 strain, which we reared under standard conditions at a temperature of 28 ± 1 °C on organic wheat flour containing 10% yeast [30]. Sex of pupae was determined by examining the structural differences in their genital papillae [30]. Immediately upon emergence, we staged the adults, categorizing those with untanned cuticles as 0 h post-emergence. We maintained male and female adults separately under the same conditions described above.

### 2.2. Sequence Alignment

We obtained Punt sequences from both *D. melanogaster* and *T. castaneum* through the National Center for Biotechnology Information (https://www.ncbi.nlm.nih.gov/ accessed on 12 March 2023). We performed sequences alignment using Clustal W2 (https://www.ebi.ac.uk/Tools/msa/clustalw2/ accessed on 13 March 2023). To visualize and annotate the resulting alignment, we utilized Jalview software version 2.11.1.4. [31].

### 2.3. Temporal and Spatial Expression Sample Preparation

We collected samples from pools of multiple individuals across various developmental stages to analyze the transcript level of *Punt*. Specifically, we collected 0.05 g of 1- and 5-day eggs, 1-, 5-, and 10-day larvae, and three individuals for each of 15-, 18-, 19-, and 20-day larvae. We also collected three individuals for each of 1–5-day female pupae and 0.5, 1 h, 1-, and 3-day female adults for analysis.

To analyze the spatial transcript levels of *Punt*, we isolated the central nervous system (CNS), fat body, gut, hemocyte, epidermis and Malpighian tubule (MT) from 90 18-day larvae. Additionally, we collected the CNS, fat body, gut, hemocyte, epidermis, MT and ovary from 90 1-day female adults. For tissue isolation, live larvae or adults were transferred to a sterile dissecting dish containing Ringer’s solution (130 mM NaCl, 4 mM KCl, 3 mM CaCl_2_, 12 mM NaHCO_3_, pH 7.4) supplemented with 1 unit/μL of RNase inhibitor. With the use of forceps and fine scissors, we carefully removed the outer exoskeleton and any undesired tissues under a dissecting microscope (Wild Heerbrugg WILD M5A Stereo Scope, Adlon Instruments Inc., St. Louis, MO, USA) to expose the specific target area, such as the CNS, fat body, gut, hemocytes, epidermis, MT, and ovary. We conducted three biological replicates for each treatment.

### 2.4. Double-Stranded RNA (dsRNA) Synthesis and RNAi Assay

To synthesize dsRNA templates for GFP, Punt and Relish, we followed the previous procedure [11], performed PCR using gene-specific primers that include the T7 polymerase promotor sequence at their 5′ ends (Appendix A), and purified the resulting cDNA using a PCR clean-up kit (M1001-50, EZ BioResearch LLC, St. Louis, MO, USA). The dsRNA was synthesized from the purified PCR product using the HiScribe™ T7 Quick High Yield RNA Synthesis Kit (E2050, New England Biolabs Inc., Beverly, MA, USA) following the manufacturer’s instructions. The synthesized dsRNA was purified using phenol/chloroform extraction and isopropanol precipitation, followed by dissolution in diethylpyrocarbonate-treated water [32]. To determine the concentration of dsRNA, a Nanodrop 2000 spectrophotometer (Thermo Fisher Scientific Inc., Waltham, MA, USA) was used.

The Nanoject II Auto-Nanoliter Injector (Drummond Scientific Co., Broomall, PA, USA) equipped with a 3.5-inch glass capillary tube pulled by a needle puller (Model P-2000, Sutter Instruments Co., Novato, CA, USA) was used to inject dsRNA. In each 18-day larva, we injected 50 nL of dsRNA at the concentrations of 1, 2, 4, and 8 ng/nL into the dorsal side between the 8th and 9th abdominal segments. Similarly, in each newly formed pupa, we injected 50 nL of dsRNA at the same concentrations as described above into the ventral side between the first and second abdominal segments. We used dsGFP as a control. The injected pupae were raised under standard laboratory conditions until they were ready for use.

### 2.5. RNA Extraction, cDNA Synthesis and qRT-PCR

We extracted total RNA from the whole body or tissues of staged beetles using TRIzol reagent (Thermo Fisher Scientific Inc., Waltham, MA, USA), followed by treatment with DNase I (Thermo Fisher Scientific Inc., Waltham, MA, USA) to remove any DNA concentrations. Next, we synthesized cDNA from 1 μg of total RNA using a High-Capacity cDNA Reverse Transcription Kit (Thermo Fisher Scientific Inc., Waltham, MA, USA) with RNase inhibitor, following the manufacturer’s instruction.

We used the QuantStudio 3 Real-Time PCR System (Thermo Fisher Scientific Inc., Waltham, MA, USA) for quantitative real-time PCR (qRT-PCR) as previously described [11]. The qRT-PCR reaction mixture consisted of 1 μL each of forward and reverse sequence-specific primers with a concentration of 10 pmol/μL (Appendix A), 1 μL of cDNA with concentration of 100 ng/μL, 5 μL of iTaq™ Universal SYBR^®^ Green Supermix (Biorad Laboratories, Hercules, CA, USA) and 3 μL of ddH_2_O. The qRT-PCR program comprised an initial denaturation at 95 °C for 3 min, followed by 45 cycles of 95 °C for 10 s, 60 °C for 20 s, and 72 °C for 30 s. Additionally, the melting curve was generated by gradually increasing the temperature from 65 °C to 95 °C at a 0.5 °C increment for 2–5 s. We quantified the relative levels of mRNAs in triplicate and normalized the data using *T. castaneum ribosomal protein S3* (*Tcrp3*) [33] as an internal control. The primer sequences for target genes are listed in Appendix A. We calculated the relative expression levels of genes using the 2^−ΔΔCT^ method [34] and performed three biological replicates to measure mRNA levels via qRT-PCR.

### 2.6. Bacterial Preparation and Inhibition Assay

For *Escherichia coli* incubation with larval hemolymph samples, we cultured *E. coli* (DH5α strain, Thermo Fisher Scientific Inc., Waltham, MA, USA) overnight in Luria–Bertani (LB) broth (0.5% yeast extract, 1% peptone, 1% NaCl, pH 7.0) at 37 °C with constant rotation at 200 rpm and centrifuged the culture at room temperature for 2 min at 10,000 *xg*. The resulting pellet was washed and resuspended with sterile phosphate-buffered saline (PBS) (137 mM NaCl, 2.7 mM KCl, 10 mM Na_2_HPO_4_, and 1.8 mM KH_2_PO_4_, pH 7.4). We divided selected larvae into two groups, each containing 90 larvae (30 for each dose) injected with dsPunt or dsGFP (100, 200 and 400 ng/larva). At 72 h post injection, we collected larval hemolymph for antibacterial activity assays, as previously described [35]. We then mixed hemolymph from 30 larvae with *E. coli* (10^3^ cells) and incubated for 6 h at 37 °C. The mixture was serially diluted, plated on LB medium, and cultured at 37 °C for 24 h. We photographed and counted the colonies on each plate. Each experiment was carried out in triplicate.

### 2.7. Measurement of Survival Rate

We divided the *T. castaneum* colony into four groups, each consisting of 90 larvae, to evaluate the larval survival rate to *E. coli* challenge after knocking down *Punt*. We used dsGFP and PBS treatment as a control. At 72 h post dsPunt injection, we injected the larvae with 50 nL *E. coli* (5 × 10^5^ cells/larva, a predetermined LC_50_ concentration). The number of dead larvae was monitored every 24 h, and the survival rates were calculated. All experiments were carried out in triplicate.

### 2.8. Mating Assays

We followed a previously established protocol [36], wherein individual females were injected with dsRNA 7 days after emergence when the ovary had matured. These females were then mated with untreated virgin males in separate tubes (24 × 62 mm, 15 mL) containing wheat flour and yeast. Each tube was stocked with a pair of mating beetles, and the beetles were transferred to fresh tubes every 24 h for egg-laying. We determined female fecundity by counting the number of eggs laid by each female per day. We calculated the hatch rate by dividing the number of hatched larvae by the total number of eggs laid.

### 2.9. Phenotype Observation/Change after RNAi

We collected female adults that were 7–9 days post emergence separately after injecting them with either dsPunt or dsGFP. We then dissected the ovaries and ovarioles from the dsRNA-treated and control specimens and placed them in Ringer’s solution (130 mM NaCl, 4 mM KCl, 3 mM CaCl_2_, 12 mM NaHCO_3_, pH 7.4). Subsequently, we photographed the specimens using a Leica M205 C stereomicroscope with a digital camera (Leica Microsystems, Germany). Additionally, we used the same equipment to photograph adults from different dsRNA-treated groups.

### 2.10. Statistical Analysis

We performed one-way analysis of variance (ANOVA) to compare gene transcript levels and other parameters between the dsPunt- or dsRelish-treated and control groups and used unpaired *t*-test to compare microbial colonies, survival rate, egg production, egg hatch rate, ovarian, and ovariole size between dsPunt-treated and control groups. In addition, we employed the graphic software Prism (Graph Pad Software, v8.1.2, San Diego, CA, USA) for figure drawing. We collected all data from three or more independent experiments and presented the results as mean ± standard error of the mean (SEM). A *p*-value < 0.05 was considered statistically significant.

## 3. Results

### 3.1. Sequence Alignment and Developmental and Tissue-Specific Expression Profiles of Punt

*T. castaneum* Punt (TcPunt; EEZ97734.1) was compared with Punt from *D. melanogaster* (DmPunt; NP_731926.1). TcPunt shares the high identity (50.6%) with DmPunt (Figure 1A).

*Punt* was expressed through all developmental stages, with the highest transcript level in 1-day female pupae, followed by 0.5 h female adults, and the lowest transcript level in 3-day female adult, followed by 1-day eggs and 18-day larvae (Figure 1B). *Punt* was constitutively expressed in the indicated tissues and organs prepared from 18-day larvae with the highest transcript level in MT followed by CNS (Figure 1C), whereas in 1-day females, ovary contained relatively high level of *Punt* transcript (Figure 1D). These suggested that Punt might function differently in adults and larvae, and possibly play a vital role in reproduction. The 18-day larvae were selected for dsPunt injection in the following experiment since *Punt* transcript level was low at this larval stage, which allows RNAi to function more effectively.

### 3.2. Punt RNAi in Larvae Increased Transcript Levels of AMP Genes

With the increase in dsPunt injection doses from 50 to 400 ng per larva, the RNAi efficiency gradually rose from 22.5% to 68.3% at 72 h post-injection (Figure 2A). To evaluate the temporal dsPunt RNAi efficiency, samples were collected from larvae injected with 200 ng/larva at 12, 24, 48, and 72 h. Figure 2B revealed that *Punt* transcript levels significantly decreased from 24 to 72 h post-injection, with RNAi efficiency increasing from 44.8% at 24 h to 70.2% at 72 h compared to the control group injected with the same dose of dsGFP. Interestingly, the transcript levels of AMP genes, *Attacin 1* (*Att1*), *Att2*, *Coleoptericin 1* (*Col1)*, *Defensin 2* (*Def2*), and *Def3*, increased by 2.5–5.1-fold at 72 h post dsPunt injection compared to the dsGFP group (Figure 2C). Notably, the transcript levels of *Att1*, *Att2* and *Def2* had a significant rise at 48 h post dsPunt injection compared with control (Figure 2C). However, the transcript level of another AMP gene, *Cecropin 2* (*Cec2*), did not exhibit a statistically significant change at 48 or 72 h after dsPunt injection. Nevertheless, it is noteworthy that there was an observable increasing trend in the transcript levels of *Cec2*, suggesting a potential modulation of expression in response to the dsPunt treatment (Figure 2C). These results suggest that *Punt* RNAi in 18-day larvae is able to induce the transcript levels of AMP genes.

### 3.3. Punt RNAi Increased Larval Anti-E. coli Activity

Furthermore, the survival rate of *Punt* RNAi larvae infected with *E. coli* increased significantly compared to the control group from 2 to 6 days post *E. coli* injection (Figure 3A). At 6 days, only 48.5% of the beetles survived in the dsGFP control group injected with *E. coli*, while 87.0% of the beetles injected with *E. coli* after *Punt* knockdown remained alive. The dead pupae in the control group injected with *E. coli* were dark brown with broken cuticles, while the newly emerged pupae in the *Punt* RNAi group injected with *E. coli* appeared normal (Figure 3A).

Hemolymph samples were collected from larvae 72 h post-injection of different doses of dsPunt or dsGFP (control) into 18-day larvae and incubated with *E. coli* (10^3^ cells) at 37 °C for 6 h. The number of *E. coli* colonies in the hemolymph sample collected from larvae injected with 100 ng/larva dsPunt decreased by 78.7% compared to the dsGFP control group. The bacterial inhibitory effect increased with the increased doses of dsPunt, reaching 52.5% and 56.3% inhibition at a dose of 200 ng/larva and 400 ng/larva, respectively (Figure 3B).

### 3.4. Punt RNAi Induced AMP Transcript Levels through Relish

Since Relish is the NF-κB transcription factor specifically functioning in the IMD pathway, which is typically activated by Gram-negative bacteria like *E. coli*, we investigated whether the induction of AMP transcript levels stimulated by dsPunt treatment works through Relish. dsRelish (400 ng/larva) was injected into 16-day larvae and 48 h later the RNAi efficiency was 65.6% (Figure 4A). Then, larvae were injected with dsPunt or dsGFP and collected 72 h post treatment for evaluating the effect of dsRelish treatment on dsPunt induced AMP transcript levels. As shown in Figure 4B, *Punt* RNAi induced the transcript levels of two representative AMP genes *Att1* and *Col1* in dsGFP control larvae. However, this enhancement did not occur in the larvae injected with dsRelish, indicating that *Punt* RNAi induced the AMP gene expression through Relish.

### 3.5. Punt RNAi Caused Adult Elytra Splitting and Abnormal Compound Eyes

It was observed that 46.1% of resulting adult beetles experienced elytra split after dsPunt injection, whereas in the control group, only 3.3% adults had this phenotype (Figure 5A). In addition, 37.9% of beetles showed abnormal compound eye color (Figure 5B).

### 3.6. Punt RNAi in Female Pupae Induced AMP Gene Expression and Impaired Female Fecundity

Since *Punt* showed high expression in 1-day female pupae and 1-day female ovaries, indicating that Punt might be involved in reproduction, we injected dsPunt (200 ng/pupa) into newly formed female pupae. The *Punt* RNAi efficiency reached 59.7% in the 3-day adults and 79.8% in the 5-day adults (Figure 6A). Following the *Punt* RNAi, transcript levels of *Att2*, *Def2*, and *Cec2* significantly increased by 1.8-, 3.1-, and 1.7-fold, respectively in the 3-day adults, and by 3.5-, 2.1-, and 2.2-fold in the 5-day adults compared to the control (Figure 6B). However, transcript levels of other AMP genes, *Att1*, *Col1*, and *Def3*, remained unaffected following dsPunt injection in female pupae compared with the control groups (Figure 6B).

The potential role of Punt in reproduction was further explored by evaluating the fecundity and egg hatch rate in females treated with *Punt* RNAi or control females mated with untreated males. Compared with the average of 9.1 eggs laid per female per day in the dsGFP control group, the average of eggs laid in the dsPunt-treated group was 2.4, down by 73.6% (Figure 7A). Furthermore, the egg hatching rate in the dsPunt-treated females was zero, indicating that there was no successful hatching (Figure 7B), while the average hatch rate in the dsGFP-treated group was 92.2%. The decline in fecundity caused by *Punt* RNAi might be related to the development of ovaries, so we dissected ovaries and ovarioles from the 7-, 8-, and 9-day female beetles after *Punt* knockdown. Ovaries from the *Punt* RNAi female could not mature 7 days after eclosion while control female ovaries were able to mature at this time point. Compared with normal ovaries (average width: 1.07 µm, average length: 1.24 µm) and ovarioles (average length: 1.13 µm), *Punt* RNAi female ovaries (average width: 0.60 µm, average length: 0.82 µm) and ovarioles (average length: 0.72 µm) were much smaller (Figure 7C,D and Appendix A). The ovaries from 8- and 9-day female were still not fully developed, although their sizes were partially recovered (Figure 7C,D and Appendix A).

## 4. Discussion

The objective of this study was to investigate the impact of knocking down TGF-β Type II receptor *Punt* on the innate immunity and development of *T. castaneum*. First, we determined the developmental and tissue-specific distribution of *Punt* transcript level. Then, we found knockdown of *Punt* in larvae significantly increased the transcript levels of AMP genes via Relish in the IMD pathway, leading to enhanced antimicrobial activity. Thirdly, we observed abnormal elytra and compound eye pigment in adult beetles following *Punt* knockdown in the larval stage. Finally, we found that *Punt* RNAi during female pupal stage resulted in the upregulation of AMP transcript levels as well as impaired ovarian development and fecundity in resulting female adults.

*Punt* transcripts were detected throughout development stages, with the prominent expression in 1-day female pupae. Additionally, transcript levels of *Punt* were abundant in CNS, MT and ovaries. Furthermore, after knocking down *Punt* in the 18-day larvae, the transcript levels of AMP genes (*Att1*, *Att2*, *Col1*, *Def2* and *Def3*) were upregulated 72 h post dsPunt injection compared with control group. These results suggest that loss of *Punt* expression stimulates the larval innate immune response; in other words, Punt is a suppressor of innate immunity. The hypothesis was confirmed as *Punt* RNAi treatment resulted in an enhanced ability of larval hemolymph to inhibit the proliferation of *E. coli*, potentially attributed to the upregulation of AMPs in the hemolymph. Furthermore, the larval survival rate significantly increased when they were injected with the LC_50_ concentration of *E. coli* after *Punt* RNAi treatment, in comparison to the dsGFP-treated control group. TGF-β is known to contribute to systemic immune suppression and inhibit host immunosurveillance in vertebrates [37]. Similarly, in *Drosophila*, Punt is involved in the inhibition of a specific arm of the immune response by both Dpp and Daw. Following injury, Dpp is rapidly activated and suppresses the production of AMPs. In *Drosophila* with dysfunctional Dpp, expression levels of AMP genes remain persistently high even after minor injuries [4]. The similar is true in the present study: knockdown of *Punt* also induced the AMP gene expression in *T. castaneum* larvae and adults. Knockdown of *Relish* eliminated the transcript levels of two AMP genes, *Att1* and *Col1*, enhanced by *Punt* RNAi, indicating that Relish is a crucial factor in the immune response triggered by *Punt* RNAi. Together, our results suggest that knocking down *Punt* enhances the innate immunity of *T. castaneum* in the larval and adult stages.

In addition to affecting the immune response of larvae, dsPunt injection in the larval stage also changed the characteristics of resulting adults, with a significant proportion of adults showing split elytra and whiten compound eyes. Elytra, the heavily sclerotized beetle forewing, serve as protective covers against mechanical stress, dehydration, and predation. Previous research in *D. melanogaster* has indicated changes in *Punt* transcript levels in adults are strongly associated with wing size reduction and loss of veins [38]. By analogy to *D. melanogaster*, our results show that alternations in the transcript level of *Punt* also affect the elytra development of adults in *T. castaneum*. As for the whitened compound eyes caused by *Punt* RNAi, we speculated that Punt might be involved in compound eye pigment patterning, which is of great importance for insect behavior, survival, and adaptation to the environment [39]. However, the underlying mechanism needs further study.

After knocking down *Punt* during female pupal stage, the ovarian and oocyte development in the resulting female adults were abnormal, leading to significantly reduced fecundity as evidenced by a decrease in egg production and hatching rate. In *Drosophila*, Dpp plays a critical role in normal oogenesis and serves as a proliferation signal for germline stem cells [17,40]. However, the exact function of TGF-β in reproduction for other insect species remains unclear, as TGF-β signaling has different regulatory functions during various periods of insect development. In humans, TGF-β signaling is complex and plays paradoxical roles in both tumor suppression and promotion, with ligands and receptors functioning differentially during different stages [41,42]. Similarity, in *Drosophila*, Dpp signaling stimulates the synthesis of amnioserosa, which is a significant source of 20E, during embryogenesis, whereas in larval stage, Dpp signaling regulates the synthesis of JH by activating JH acid methyltransferase, a key enzyme in the JH biosynthesis pathway [43,44]. As insects grow and develop, the levels of JH gradually decrease, while the levels of 20E increase [45]. Combined with these studies, TGF-β signaling may have different regulatory functions in various periods during the growth and development of *T. castaneum*. To further understand the biological role of Punt in TGF-β signaling of insects, the specific effect of *Punt* RNAi on Relish at the protein level needs further study, since when activated, Relish undergoes cleavage into two fragments: the N-terminal DNA binding fragment migrates into the nucleus, while the C-terminal IκB-like fragment remains stable in the cytosol [29,46]. Furthermore, the impact on ovarian and oocyte development in females requires more detailed mechanistic studies, such as whether *Punt* RNAi affects JH or 20E titers to further impact reproduction. Regardless, it is not surprising to see that enhanced immunity after *Punt* knockdown leads to reduced reproduction. The transcript levels of AMP genes were upregulated following dsPunt injection in female pupae, although the effect of *Punt* RNAi on AMP genes varied slightly between the larval and pupal stages. In larvae, the transcript level of *Cec2* showed an upward increasing trend, although not statistically significant, following dsPunt injection, but it was significantly increased when dsPunt was injected into female pupae. This could be attributed to the possibility that TGF-β signaling functions differently across various developmental stages of *T. castaneum.* Most importantly, it has been well documented that there is a tradeoff between immunity and reproduction in a diversity of female insects. Increased immunity by infection and activation of the immune system results in reduced reproduction, and reciprocally, increased reproductive output reduces immunity [47]. Still, this study offers valuable insights into the biological importance of Punt in insect TGF-β signaling, establishing a basis for future research on the role of insect TGF-β signaling in regulating innate immunity, development, and reproduction.

## Figures and Tables

**Figure 1 insects-14-00515-f001:**
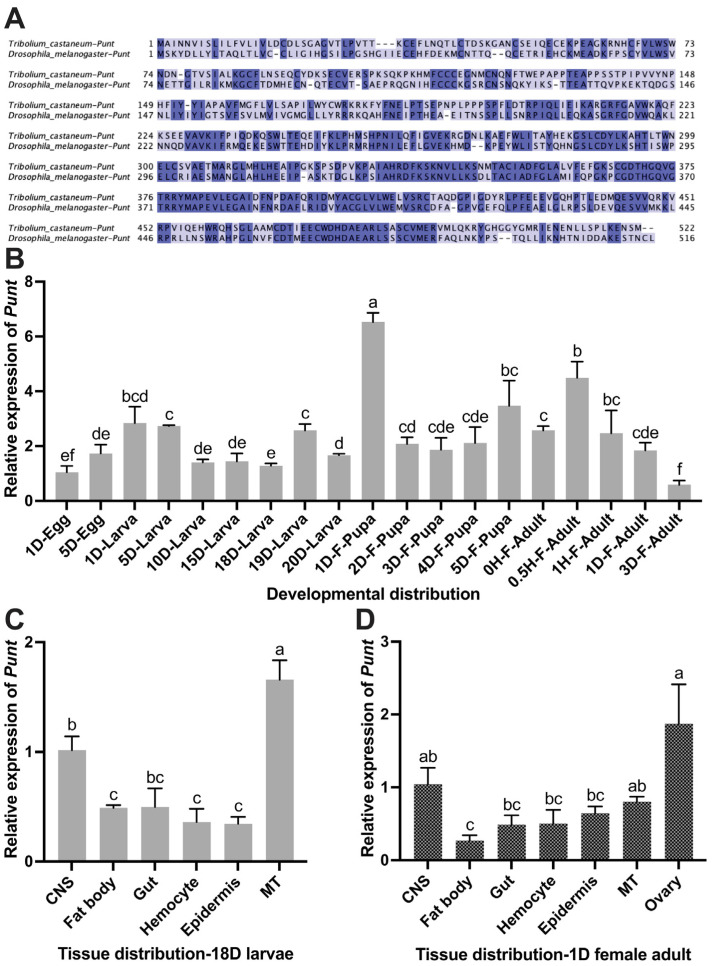
Sequence analysis, developmental and tissue-specific distribution of *Punt*. (**A**) Double-sequence alignment of Punt amino acid sequences from *T. castaneum* and *D. melanogaster*. Conserved amino acid sequences are indicated by a dark blue background. Identical and highly similar amino acid sequences are indicated by a light blue background. (**B**) Developmental profiles of *Punt* transcript levels. The relative transcript levels of the target transcripts in the 1-day egg served as the calibrator for the developmental expression profiling. (**C**) Tissue-specific transcript levels of *Punt* in central nervous system (CNS), fat body, gut, epidermis, hemocyte and Malpighian tubule (MT) from 18-day larvae. (**D**) Tissue-specific transcript levels of *Punt* in CNS, fat body, gut, epidermis, hemocyte, MT and ovary from 1-day female adults. Different letters on the top of bars indicate that the means ± SEM are significantly different between treatments by *t*-test (*p* < 0.05).

**Figure 2 insects-14-00515-f002:**
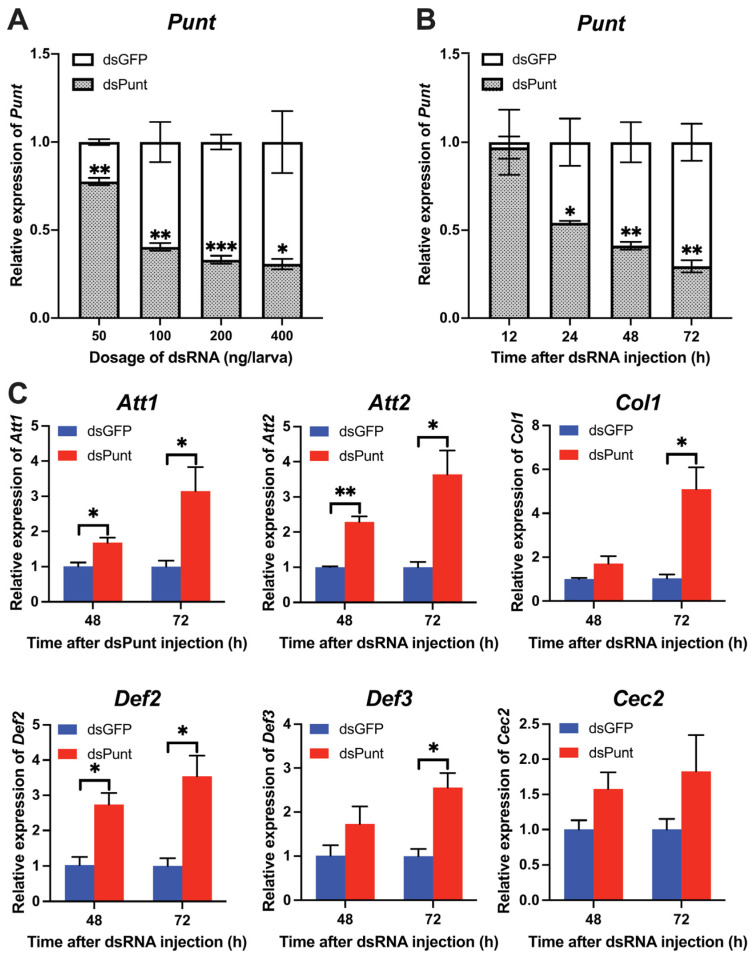
The efficiency of *Punt* RNAi in larvae and its effect on AMP gene transcript levels. (**A**) Relative transcript levels of *Punt* 72 h post injection with different doses of dsPunt or dsGFP. (**B**) Relative transcript levels of *Punt* 12–72 h post injection with dsPunt or dsGFP at 200 ng/larva. (**C**) Transcript levels of AMP genes in 18-day larvae 48 and 72 h post injection of dsPunt and dsGFP at 200 ng/larva, respectively. Asterisks above bars indicate significant differences between the treatment and corresponding control, * *p* < 0.05, ** *p* < 0.01, *** *p* < 0.001 by *t*-test.

**Figure 3 insects-14-00515-f003:**
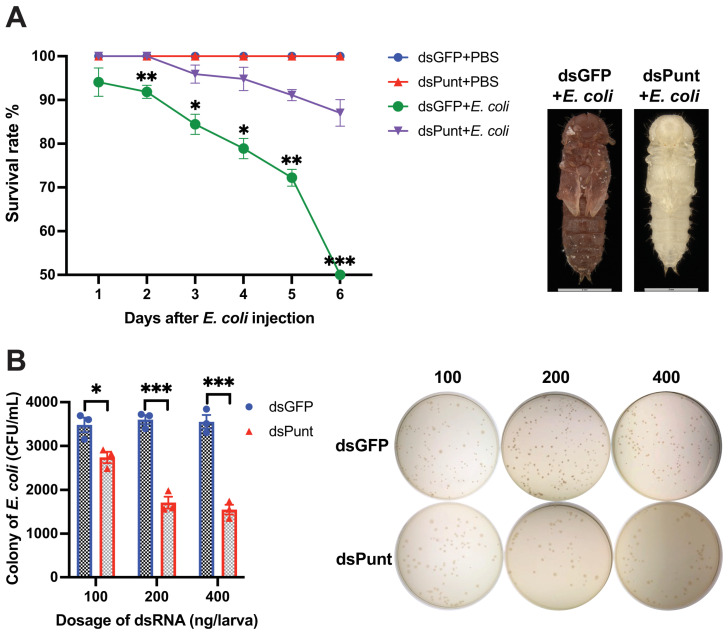
Effect of *Punt* RNAi on the larval anti-*E. coli* activity. (**A**) Survival rate of *Punt* RNAi larvae infected with *E. coli* and the resulting phenotypes of pupae from dsGFP+*E. coli* group and newly emerged pupae from dsPunt+*E. coli*. (**B**) CFU of *E. coli* after 10^3^ cells of *E. coli* incubated with hemolymph collected from larvae 72 h post injection of different doses of dsPunt or dsGFP for 6 h. Asterisks above bars indicate significant differences between the treatment and corresponding control, * *p* < 0.05, ** *p* < 0.01, *** *p* < 0.001 by *t*-test.

**Figure 4 insects-14-00515-f004:**
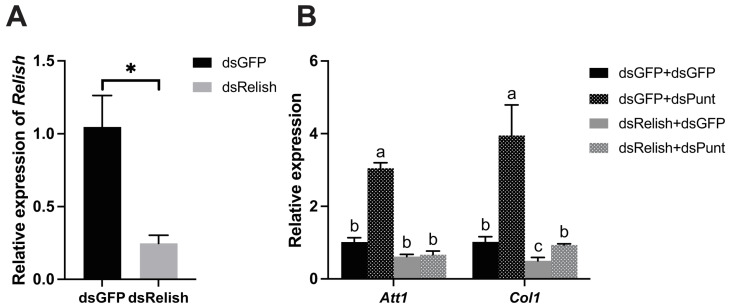
*Punt* RNAi induced AMP gene expression through Relish. (**A**) The transcript level of *Relish* in larvae 48 h post dsRelish injection. (**B**) qRT-PCR analysis of 2 representative AMP genes *Att1* and *Col1* in the dsPunt or dsGFP treated larvae for 72 h after dsGFP- and dsRelish-injection for 48 h. Asterisks above bars indicate significant differences between the treatment and corresponding control, * *p* < 0.05 by *t*-test. Different letters on the top of bars indicate that the means ± SEM are significantly different among treatments by *t*-test (*p* < 0.05).

**Figure 5 insects-14-00515-f005:**
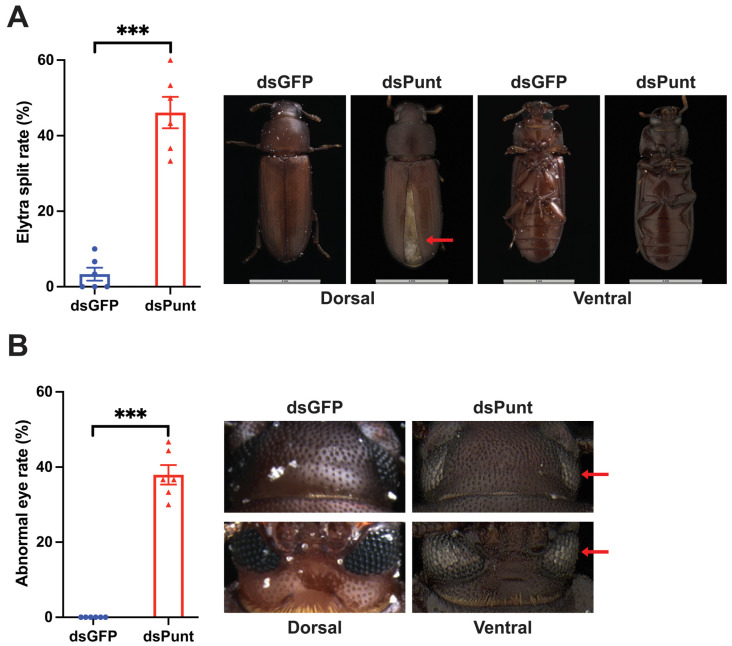
Effects of *Punt* RNAi on adults. (A) Elytra wing splitting rate after *Punt* RNAi and the phenotype of 1-day adults post *Punt* RNAi. Red arrow indicates splitting elytra. (B) Abnormal eye rate post *Punt* RNAi and comparison of 1-day adult eye colors post *Punt* RNAi. Red arrows indicate abnormal compound eye coloration. Asterisks above bars indicate significant differences between the treatment and corresponding control, *** *p* < 0.001 by *t*-test.

**Figure 6 insects-14-00515-f006:**
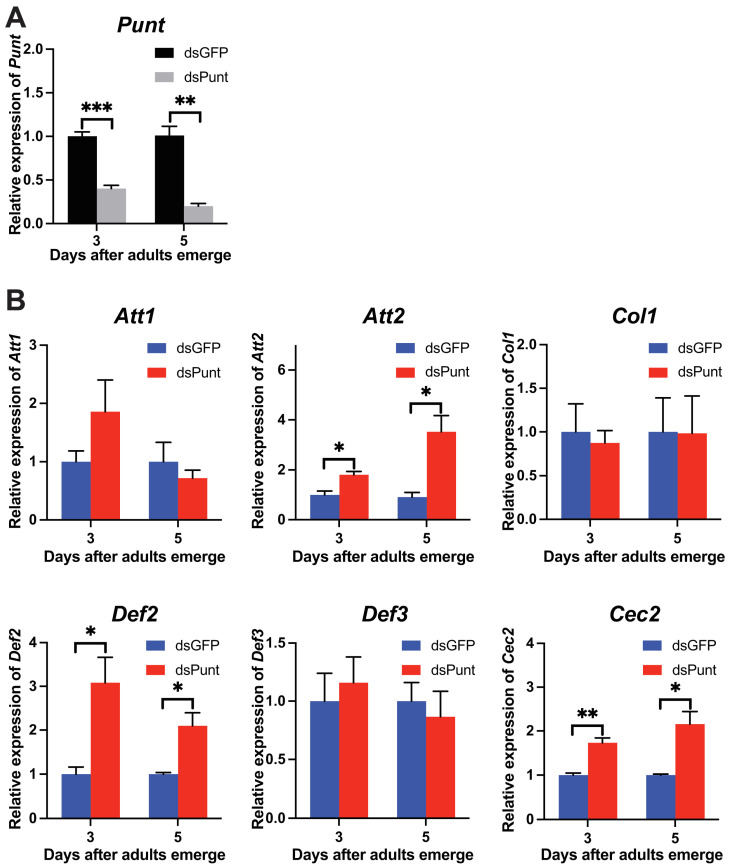
The efficiency of *Punt* RNAi in female adults and its effects on AMP gene transcript levels. (**A**) Transcript levels of *Punt* in the 3- and 5-day females after dsGFP or dsPunt (200 ng/larva) treatments in the newly formed female pupae. (**B**) Transcript levels of AMP genes in the 3- and 5-day females post injection of dsGFP or dsPunt (200 ng/larva) into newly formed female pupae. Asterisks above bars indicate significant differences between the treatment and corresponding control, * *p* < 0.05, ** *p* < 0.01, *** *p* < 0.001 by *t*-test.

**Figure 7 insects-14-00515-f007:**
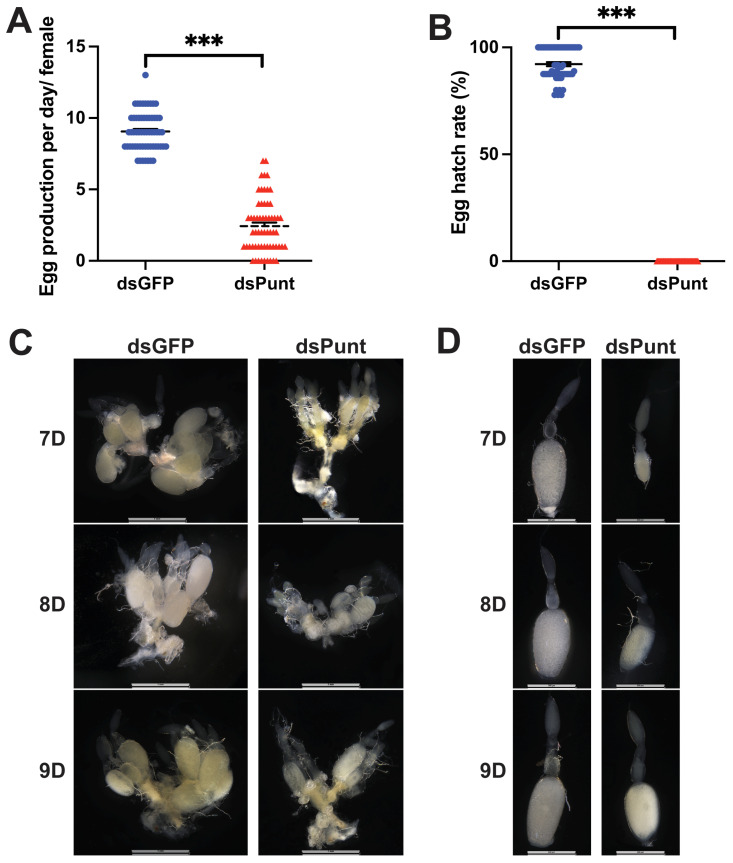
The effects of *Punt* RNAi on female reproduction. (**A**) The number of eggs laid per day per female in the dsGFP- or dsPunt-treated females. (**B**) Egg hatch rate in the dsGFP- or dsPunt-treated females after mating with untreated males. (**C**) The ovaries from the 7-, 8- and 9-day females after dsGFP or dsPunt treatments. (**D**) Ovarioles from the 7-, 8- and 9-day females after dsGFP or dsPunt treatments. Asterisks above bars indicate significant differences between the treatment and corresponding control, *** *p* < 0.001 by *t*-test.

## Data Availability

Data are contained within the article and Appendix A.

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
