# Peer review of "TGF-β Type II Receptor Punt Suppresses Antimicrobial Peptide Expression and Influences Development in Tribolium castaneum"

_insects, 2023, doi:10.3390/insects14060515_

Round 1
Reviewer 1 Report
1. The results showed that knockdown of Punt in larvae caused splitting of adult elytra and abnormal compound eyes, and knocking down Punt during the female pupal stage resulted in increased transcript levels of AMP genes as well as abnormal ovarian development, decreased fecundity, and failure of egg hatching. The results showed that Punt can affect the development of T. castaneum, and affect the antimicrobial peptide (AMP) expression but can’t reveal its role in mediating the immunity progress. So, it will be exactly for the title to focus on “affecting development” not only “suppresses antimicrobial peptide expression”
2. Introduction: the contents about TGF-β should be introduced something in Coleoptera insects, not only about Bombyx mori or Drosophila. And something about Tribolium castaneum should be introduced.
3. L378-379: “The objective of this study was to investigate the impact of knocking down TGF-β Type II receptor Punt on the innate immunity of T. castaneum” is not exactly, and it may be more accurate if it is changed to “The objective of this study was to investigate the impact of knocking down TGF-β Type II receptor Punt on the development of T. castaneum”
4. L136-139: How to isolated the central nervous system (CNS), fat body, gut, hemocyte, epidermis and Malpighian tubule (MT), and give the methods/protocol please.
5. L208: “Ovary and beetle phenotypes after RNAi” ovary is one of beetle phenotypes, so it should be “phenotypes observation/change after RNAi”
6. L241-251: What is the statistical difference for the samples? e.g. P<0.05? The P value should also be labeled in Figure 1.
Author Response
Dear Reviewer,
We sincerely appreciate the time you have taken to review our manuscript and provide valuable comments. Your input has been incredibly helpful in improving the quality of our work. We have carefully considered your suggestions and advice, and as a result, we have modified the manuscript accordingly.
Please see the attachment. We have diligently highlighted all the revisions and additions in the attached document using the red font. This document contains a detailed response addressing your comments and advice, as well as the revised manuscript itself.
Thank you very much!
Sincerely,
Dr. Qisheng Song

Reviewer 2 Report
The manuscript entitled "TGF-beta type II receptor suppress antimicrobial expression in Tribolium castaneum" represents a well balanced experimental approach allowing most of the firm conclusion statements.
Some suggestion might be in place in order to further improev the manuscript prior to publication.
1) Line 20. Do authors really mean 'insects' here as it feels they mean vertebrates here ( see line 23-24: However... in insects..). Why in line 20 use of past sentence' Regulated'??
2) Throughout text the author use dsGFP and dsPUNT whereas they mean dsGFP RNA and dsPUNT RNA.
M&M line 187. The way of co-incubating hemolymp and E.coli will give information about the presence of bacteriostatic/ bacteriocidal compounds in the hemolymph sample. However in the discussin section line 394 the authors refer to "ability to clear E. coli". The latter refers to cellular immune activity either by hemocytes/ cardiac tissue or fatbody cells which is not included in this study.
Results: lin 264-265 As fig 2C celarly show an increase (though not being significant) both at 48hrs and 72hrs the firm statement of PUNT RNAI not affecting CEC2 is incorrect. Please weaken conclusion! See also Line 442 wher authors incorrectly mention the lack of CEC2 being affected by PUNT RNAi!
Fig 5B the photo's of the eye development are rather unclear.
Discussion:
see above statements regarding E.coli assay ( line 187 and 394)
see above statement regarding CEC2 not being affected by PUNT RNAi ( lines 246 and 442)
Author Response
Dear Reviewer,
We would like to express our sincere gratitude for taking the time to review our manuscript and providing valuable feedback. Your insightful comments have played a crucial role in enhancing the quality of our work. We have thoroughly considered your suggestions and advice, so we have made appropriate modifications to the manuscript.
Please see the attachment. To assist you in the review process, we have diligently highlighted all the revisions and additions in the attached document using red font. This document contains a detailed response addressing your comments and advice, as well as the revised manuscript.
Once again, we extend our heartfelt thanks for your invaluable contribution to our research. Your expertise and guidance have greatly enriched our work.
Thank you very much!
Sincerely,
Dr. Qisheng Song
